# SynToxProfiler: An interactive analysis of drug combination synergy, toxicity and efficacy

**Aleksandr Ianevski[1,2], Sanna Timonen[1], Alexander Kononov[1], Tero Aittokallio[1,2,3]\*, Anil K. Giri[1]\***

**1** Institute for Molecular Medicine Finland (FIMM), University of Helsinki, Helsinki, Finland, **2** Helsinki Institute for Information Technology (HIIT), Aalto University, Espoo, Finland, **3** Department of Mathematics and Statistics, University of Turku, Quantum, Turku, Finland

\* tero.aittokallio@helsinki.fi (TA); anil.kumar@helsinki.fi (AKG)

**Data Availability Statement:** All the data are available at the website and technical documentation.

**Funding:** This work was supported by Academy of Finland (grants 292611, 279163, 295504, 310507,

## Abstract

Drug combinations are becoming a standard treatment of many complex diseases due to their capability to overcome resistance to monotherapy. In the current preclinical drug combination screening, the top combinations for further study are often selected based on synergy alone, without considering the combination efficacy and toxicity effects, even though these are critical determinants for the clinical success of a therapy. To promote the prioritization of drug combinations based on integrated analysis of synergy, efficacy and toxicity profiles, we implemented a web-based open-source tool, SynToxProfiler (Synergy-Toxicity-Profiler). When applied to 20 anti-cancer drug combinations tested both in healthy control and T-cell prolymphocytic leukemia (T-PLL) patient cells, as well as to 77 anti-viral drug pairs tested in Huh7 liver cell line with and without Ebola virus infection, SynToxProfiler prioritized as top hits those synergistic drug pairs that showed higher selective efficacy (difference between efficacy and toxicity), which offers an improved likelihood for clinical success.

## Author summary

High-throughput combinatorial screening is an established approach to identify candidate drug combinations to be further developed as safe and effective treatment options for many diseases, such as various types of cancers, bacterial, malarial, and viral infections. The selection of top performing drug combinations for further development is an important step for the success of the screen, where not only the synergy but also selective efficacy and potential toxicity of the drug pairs should be critically assessed. Currently, there is no method available for this; therefore, we developed SynToxProfiler tool, which was demonstrated in two different application cases to prioritize synergistic drug pairs with higher efficacy and lower toxicity as top hits, providing thus an increased likelihood for their clinical success.

326238), European Union's Horizon 2020 Research and Innovation Programme (ERA PerMed JAKSTAT-TARGET), the Cancer Society of Finland (TA) and the Sigrid Jusélius Foundation (TA). The funders had no role in study design, data collection and analysis, decision to publish, or preparation of the manuscript.

**Competing interests:** The authors have declared that no competing interests exist.

## Introduction

High throughput screening (HTS) of approved and investigational agents in preclinical model systems has been established as an efficient technique to identify candidate drug combinations to be further developed as safe and effective treatment options for many diseases, such as HIV, tuberculosis and various types of cancers [1, 2]. Currently, the selection of top combinations for further development often relies merely on the observed synergy between drugs, while neglecting their actual efficacy and potential toxic effects, that are the other key determinants for the therapeutic success of drugs in the clinics [3]. Notably, around 20% of drugs fail in the early development phase because of safety concerns (non-tolerated toxicity), and over 50% fail due to lack of sufficient efficacy [4]. Further, a recent study argued that many clinically-used anticancer combination therapies confer benefit simply due to patient-to-patient variability, not because of drug additivity or synergy [3], indicating that even non-synergistic combinations may be beneficial for therapeutic purposes if they have a high enough efficacy and low enough toxicity profiles.

To make a better use of these various components of drug combination performance already in the preclinical HTS experiments, we implemented, to the best of our knowledge, the first web-tool, SynToxProfiler, which enables users to profile synergy, toxicity and efficacy of drug combinations simultaneously for the top combination prioritization (Fig 1). For each drug pair, SynToxProfiler calculates the total volume under the multi-dose combinatorial response surface measured in the diseased and healthy cells, respectively, to quantify efficacy and toxicity scores (S1 Fig). Further, synergy matrix is estimated from the dose-response matrix of the diseased cells using an appropriate synergy model (e.g. Highest-single agent, Bliss, Loewe, or Zero interaction potency model (ZIP)) [5–8], depending on the drugs and hypotheses of the HTS experiment. The synergy score is calculated as the integrated volume under the estimated dose-synergy matrix. Further, each combination is ranked based on its integrated synergy, toxicity and efficacy scores. While the current implementation focuses on pairwise drug combinations, the method is also extendable for higher-order (3 or more drugs) combination screening.

In this report, we demonstrate the use of SynToxProfiler as a systematic tool for the prioritization of top combinatorial hits both in T-cell prolymphocytic leukemia (T-PLL) and anti-Ebola drug combination screening. The SynToxProfiler platform is implemented as an open-source and interactive web-tool (https://syntoxprofiler.fimm.fi), which enables fast and fully-automated top hit selections in HTS combination screening with 2D and 3D visualization (Fig 1, S2 Fig). To guarantee the replicability of the work and possibilities for further extension of the tool by academic users, we have also provided the source code of the method at https://github.com/IanevskiAleksandr/SynToxProfiler.

## Results

### SynToxProfiler prioritizes clinically useful drug combination for T-PLL cancer patient and Ebola virus infection

To demonstrate the performance of SynToxProfiler in prioritizing therapeutically-relevant synergistic combinations, we applied it to the in-house drug screening data involving 20 anti-cancer drug combinations tested in one control and one T-PLL patient-derived cells. The application of SynToxProfiler to the T-PLL screen revealed that ranking of combinations based on the STE score successfully prioritizes both effective and safe drug pairs. For example, Cytarabine-Daunorubicin pair was identified as the top hit out of the tested combinations (Table 1, S1 Data); this combination is widely used as approved induction therapy for acute myeloid leukemia treatment [9]. Ibrutinib-Navitoclax was ranked as the third-best

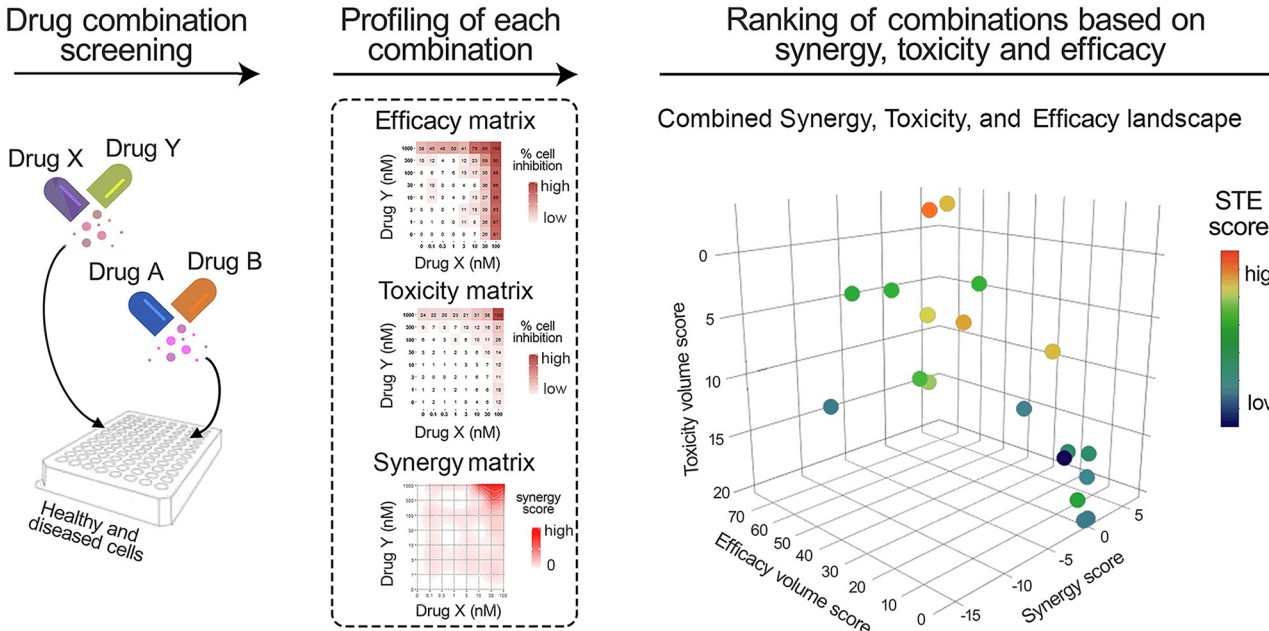

**Fig 1. A schematic overview of SynToxProfiler.** The dose-response data from drug combination screening, measured in both diseased (e.g. patient-derived cells) and healthy control cells (e.g. PBMCs), is provided as input to SynToxProfiler (left panel). Then, SynToxProfiler quantifies drug combination efficacy and synergy (using combination responses in diseased cells) as well as toxicity (using combination responses in control cells) for each drug pair (middle panel), and summarizes them into integrated synergy, toxicity and efficacy (STE) score. The increase in red color gradient in the efficacy and toxicity matrix (middle panel) denotes an increase in percentage inhibition of cancer and healthy cells, respectively. The color gradient from white to red in the synergy matrix shows the shift of drug interactions from antagonism to synergy. The STE score is further used to rank and visualize the drug pairs in 2D or 3D interactive plots (right panel). The color gradient from blue to red in the right panel depicts an increasing STE score from low to high.

combination for further study; such combinations (Ibrutinib with Bcl2 inhibitors) have shown promising results in phase II clinic trial (NCT02756897) for chronic lymphocytic leukemia (CLL), and recently suggested as the first-line treatment for CLL [10].

To further illustrate the wide applicability of SynToxProfiler also in non-cancer combinatorial screens, we used a published dataset of 77 anti-viral agents, where the drug combinations efficacy and toxicity were tested in Ebola-infected and non–virus-infected Huh7 liver cells, respectively. SynToxProfiler ranked the established combinations (e.g. clomifene-sertraline and sertraline-toremifene) that inhibit EBOV fusion to cell surface as top hits in the Ebola combination screen (S2 Data). All the three drugs (clomifene, sertraline and toremifene) have showed survival benefit in in-vivo murine Ebola virus infection model [11]. These two case studies demonstrate that SynToxProfiler enables identification of both clinically established drug pairs as top hits, as well as novel drug pairs with a translational potential.

## Top hits selected by SynToxProfiler based on integrated scoring are synergistic drug pairs with higher selective efficacy

We compared the synergy and selective efficacy levels of the top hits prioritized based either on the STE score, synergy score or selective efficacy scores, using the 77 combinations in the Ebola dataset [12]. The top combinations identified by STE scores had a notably higher selective efficacy as well as higher synergy (as shown by arrow in Fig 2A), indicating that STE score represents a proper balance between high selective efficacy and synergy. Additionally, we observed a marked overlap (65%) between the top-10% of analyzed combinations prioritized

**Table 1. Ranking of 20 in-house measured combinations based on STE scores calculated from the most synergistic area of dose-response matrix in T-PLL and healthy control cells.**

| Drug1 (concentration range in nM) | Drug2 (concentration range in nM) | Synergy score ($S_{AB}$) | Efficacy score ($E_{AB}$) | Toxicity score ($T_{AB}$) | Selective efficacy score ($E_{AB}$) | STE score |
|---|---|---|---|---|---|---|
| Cytarabine (0–100) | Daunorubicin (0–1000) | 6.70 | 58.73 | 20.86 | 37.86 | 0.83 |
| Trametinib (0–100) | S-63845 (0–25) | 4.20 | 2.00 | 12.43 | 35.97 | 0.83 |
| Ibrutinib (0–1000) | Navitoclax (0–100) | 1.53 | 71.69 | 19.92 | 51.76 | 0.83 |
| Quizartinib (0–100) | S-63845 (0–100) | 0.69 | 56.38 | 27.43 | 28.95 | 0.75 |
| Omacetaxine (0–1000) | Ipatasertib (0–1000) | 0.19 | 70.10 | 11.36 | 58.74 | 0.73 |
| Gefitinib (0–1000) | Omacetaxine (0–1000) | 0.69 | 69.36 | 17.07 | 52.28 | 0.73 |
| Clofarabine (0–1000) | Idarubicin (0–100) | 2.08 | 73.88 | 27.53 | 46.35 | 0.73 |
| Omacetaxine (0–1000) | Alpelisib (0–1000) | -0.72 | 70.87 | 10.95 | 59.92 | 0.68 |
| Clofarabine (0–1000) | Prexasertib (0–1000) | 1.59 | 19.70 | 8.90 | 10.80 | 0.68 |
| Gefitinib (0–25) | Trametinib (0–1000) | 0.40 | 10.25 | 4.51 | 5.74 | 0.55 |
| Buparlisib (0–100) | Ibrutinib (0–1000) | 0.94 | 4.45 | 1.33 | 3.12 | 0.55 |
| Ibrutinib (0–100) | Doxorubicin (0–100) | 0.84 | 9.82 | 1.73 | 8.09 | 0.53 |
| Vinorelbine (0–1000) | Clofarabine (0–1000) | -1.14 | 66.46 | 21.40 | 45.06 | 0.50 |
| Clofarabine (0–1000) | Omacetaxine (0–1000) | -4.27 | 20.34 | 0.59 | 19.75 | 0.38 |
| Dexamethasone (0–1000) | Clofarabine (0–1000) | -16.37 | 0.00 | 0.00 | 0.00 | 0.35 |
| Dasatinib (0–1000) | Ipatasertib (0–100) | 0.24 | 0.67 | 2.08 | -1.41 | 0.25 |
| Carboplatin (0–1000) | Dexamethasone (0–1000) | -0.51 | 0.00 | 0.00 | 0.00 | 0.20 |
| Ipatasertib (0–1000) | ASP3026 (0–1000) | -0.01 | 0.00 | 0.00 | 0.00 | 0.18 |
| Idarubicin (0–100) | Ibrutinib (0–100) | -0.89 | 0.69 | 2.28 | -1.59 | 0.18 |
| Trametinib (0–100) | Dasatinib (0–25) | -2.05 | 0.00 | 1.23 | -1.23 | 0.10 |

Synergy scores were calculated using ZIP [8] model (default option in SynToxProfiler).

based on STE score and synergy score, as well as based on the STE score and selective efficacy score (50% overlap), as shown in Fig 2B. In contrast, there was a smaller overlap (41%) between the top-10% hits selected based on selective efficacy and synergy scores. Further, a low Pearson correlation (r = 0.22) between selective efficacy and synergy was observed. These results indicate that synergy and selective efficacy are independent drug combination components, which cannot be used alone to prioritize potent and less toxic synergistic drug combinations.

A more detailed analysis revealed that SynToxProfiler ranks lower the toxic drug pairs despite their higher synergy. For example, in the T-PLL case, SynToxProfiler ranks 9th the combination of prexasertib-clofarabine, despite its higher synergy (4th, see Table 1), because of its low efficacy and higher toxicity, as also observed in Philadelphia-negative B-/T-ALL cell line model where addition of prexasertib increased the cytotoxicity of clofarabine [13]. Similarly, SyntoxProfiler ranked clomiphene citrate and sertraline HCl combination (STE = 0.96) as the top hit in the Ebola screen (Fig 3), despite its lower synergy as compared to more synergistic toremifene citrate and apilimod pair (STE = 0.86). This is due to a higher toxicity of the latter (13.30 vs 24.60), although both of the drug combinations have similar efficacy scores (70.88 vs 68.20). The relative lower ranking of combinations involving cilchicine and apilimod is in accordance with their observed extreme toxicity in the clinic [14,15]. These results indicate that SynToxProfiler can identify safe top hits with high selective efficacy and synergy that have increased potential for clinical success, as compared to hits selected based on synergy alone.

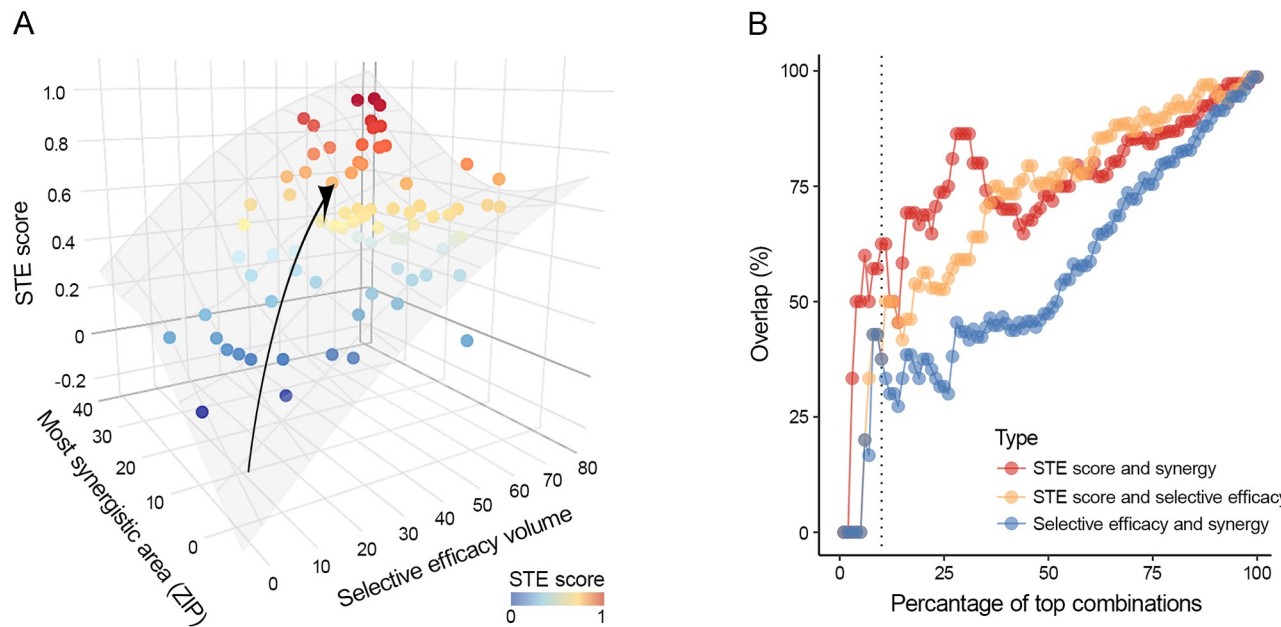

**Fig 2. STE score considers both synergy and selective efficacy when prioritizing potent drug combinations.** (A) 3D surface shows increase in STE score with increasing synergy and selective efficacy scores across 77 antiviral combinations measured in Huh7 liver cell line infected with Ebola virus (the arrow marks the gradient of the increase in STE score). The 3D surface is fitted by a generalized additive model with a tensor product smooth, implemented in mgcv R package. The color gradient from blue to red represents the low to high STE score for drug combinations. (B) Scatter plot showing the overlap in the top hits selected on the basis of STE and Synergy (red), STE and selective efficacy(orange), and selective efficacy and synergy (blue) scores. The dotted vertical line denotes the overlap between the top 10% combinations selected based on any of the three scores.

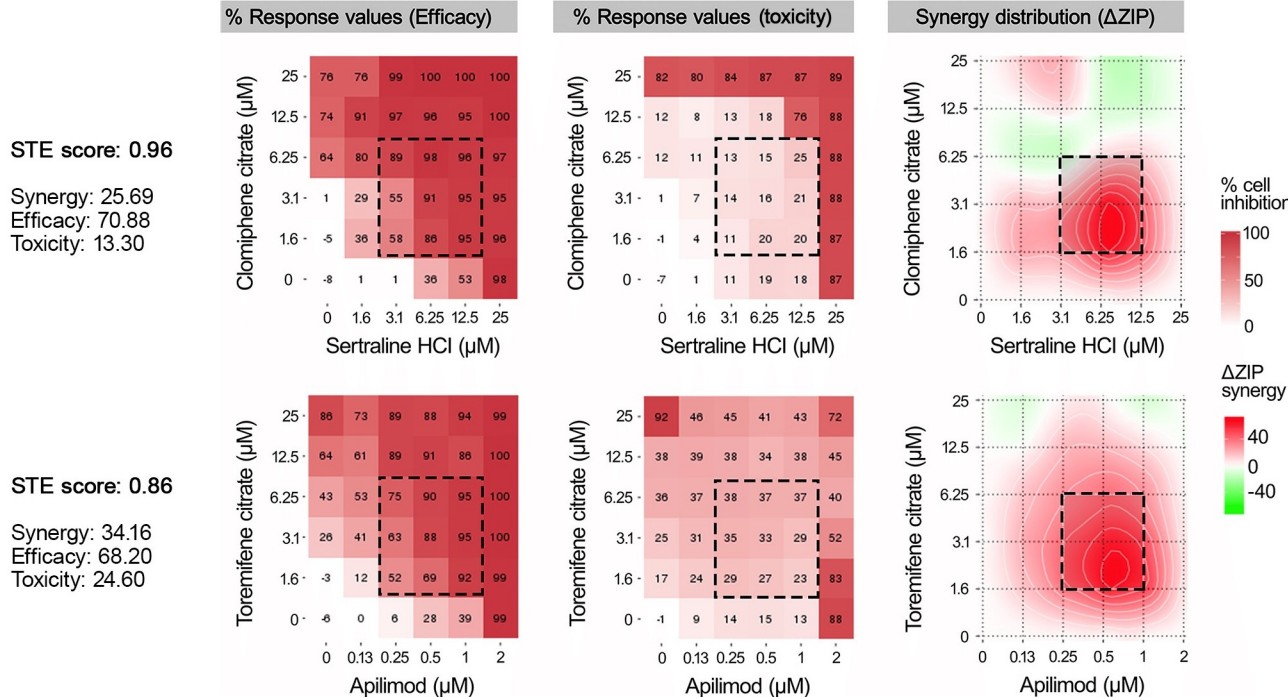

**Fig 3. SynToxProfiler penalizes for toxicity of drug pairs while ranking top hits.** The efficacy, toxicity, and synergy matrices for the top drug pairs selected based on the highest STE score (clomiphene citrate and sertraline HCL, upper panel) and the highest synergy score (toremifene citrate and apilimod, lower panel). The synergy was calculated using the ZIP model implemented in SynergyFinder [21]. The square with dotted line denotes the 3x3 concentration range with the most synergistic area in the dose-response matrix.

## Comparison of SynToxProfiler and Combenefit using different synergy scoring models

To benchmark the results of SynToxProfiler against other combination scoring methods, we firstly compared the volume-based SynToxProfiler Bliss synergy calculation against the widely-used SUM_SYN_ANT synergy score of Combenefit [16]. As expected, we observed a strong correlation in both datasets (Pearson correlation of 0.92 and 0.83 for the 20 anti-cancer and 77 anti-Ebola drug combinations, respectively) between the two methods (S3 Fig). We also observed that the top combination hits selected by SynToxProfiler are highly overlapping between different synergy models; for instance, 9 and 8 of the top-10 combinations identified based on the STE score using ZIP synergy model are actually the same when using Bliss and HSA models in T-PLL and anti-Ebola screening, respectively (S2 and S3 Tables).

Additionally, we compared the top hits from T-PLL screen ranked based on Combenefit synergy score and STE score using Bliss synergy model, and observed that STE score helps to deprioritize toxic and low efficacy drug pairs that would have been selected based on the Combenefit score. For example, Buparlisib-Ibrutinib combination was ranked 11[th] by SynToxProfiler, although it has a higher Combenefit synergy rank (6[th]), because of its lower efficacy and higher toxicity (S4 Table), similar to the observation made in Phase 1 clinical trials for hematological cancers (in relapsed/refractory diffuse large B-cell lymphoma, mantle cell lymphoma, and follicular lymphoma) [17]. Similarly, in the anti-Ebola data, SynToxProfiler ranks relatively high (4[th]) colchicine/toremifenecitrate combination, despite its low synergy rank (18[th]) given by Combenefit (S5 Table), because of its higher efficacy (S5 Table). This combination synergistically blocks Ebola infections as toremifenecitrate blocks its fusion with the cells and colchicine blocks the trafficking to endolysosomes [12], indicating that SynToxProfiler prioritizes drug pairs with a strong potential to be rapidly advanced towards clinical settings and used as therapeutic interventions.

## Discussion

The primary motivation for the use of drug combinations in the clinical applications is to achieve higher efficacy (by means of drug interactions), with reduced toxicity (by decreasing the drug doses). Therefore, HTS screening aims to discover drug pairs that are more effective than the individual single drugs when used alone, and at the same time show less toxicity for the patients. Hence, the assessment of synergistic efficacy along with toxicity is critical for the selection of candidate drug pairs for further study, as there exists a fundamental trade-off between clinical efficacy and tolerable toxicity. In this work, we showed how SynToxProfiler prioritized cytarabine-daunorubicin as the top drug pair out of 20 anticancer combinations for T-PLL case study (Table 1), and clomifene-sertraline for Ebola case study (S2 Data). The identification of such clinically established drug pairs as top hits suggests that ranking based on all the three components helps to identify combinations that have more chances to succeed in the clinical practice. Notable, these effective and safe combinations would have been otherwise missed if combinations were selected merely based on their synergy scores.

To the best of our knowledge, there are currently no methods to provide the global view in terms of synergy, efficacy and toxicity of drug pairs in an HTS setting. In this respect, SynToxProfiler offers an important advancement into the current practice for drug combination selection, as it provides an easy-to-use platform for in-vitro or ex-vivo assessment of the three critical aspects of drug combinations that are necessary for success in the clinical applications. As default, SynToxProfiler calculates the efficacy, toxicity and synergy scores within the most synergistic concentration window (3x3), which allow for the comparison of drugs at their most beneficial concentration ranges in those combination screens where the primary aim is

to find most synergistic drug pairs. Furthermore, SynToxProfiler facilitates the identification of therapeutic window range at which the drugs show the highest efficacy and lowest toxicity by visualization of the dose-response surfaces. Since SynToxProfiler uses the normalized volume-based scoring for synergy, efficacy and toxicity levels (see Methods and S1 Text), the SynToxProfiler framework can be easily utilized to prioritize synergistic drug combinations with high selective efficacy also for higher-order combination screening (3 or more drugs). Due to the lack of tools and methodology available to analyze and interpret combined synergy, efficacy and toxicity of multi-component drug combinations, SynToxProfiler will be valuable resource for prioritization of such combinations.

However, we note that although for some drug classes the *in vitro* toxicity measurement strongly corelates with the clinical toxicity, the toxicity measurements in cell lines may not accurately capture clinical toxicity for all drug classes or toxicity phenotypes [18, 19]. Hence, due to this technical limitation of *in vitro* toxicity assays, the selected combinations will need to be further tested in animal models or clinical studies. We expect that SynToxProfiler will become even more useful when toxicity measurements from biologically more relevant preclinical models, such as induced pluripotent cells and organoids of non-diseased tissue of patients, start to become available for high throughput screening [20]. However, filtering out combinations with *in vitro* toxicity should lead to savings of both time and resources, as well as to reduced animal and human suffering. SynToxprofiler can be used also to identify and characterize synergistic drug pairs with high toxicity and low efficacy in order to understand the underlying mechanism behind chemical toxicity using appropriate model system. We suggest that user should also visualize the dose-response curves of individual drugs for selected top hits, as well as the full dose-combination matrices for efficacy, toxicity and synergy estimates, to confirm the efficacy/synergy/toxicity summary scores before selection of top hits for further study. Further, we recommend that users should carefully choose the appropriate synergy model based on their underlying hypothesis behind drug interactions, as the different synergy models come with distinct assumptions for the synergy calculation. For example, one should use Loewe model when the assumption is that drugs act on the same target or binding site through the same mechanism, and Bliss model in the opposite case, when the assumption is that drugs in combination act independently through different mechanisms, but together they can exceed their individual responses.

In conclusion, we have developed SynToxProfiler, an interactive tool for HTS combination hit prioritization that ranks drug pairs based on their combined synergy, efficacy and toxicity profiles, which can be implemented in any HTS drug combination screening project, either as stand-alone tool or combined with other screening and scoring approaches. We showed how this tool enables identification of clinically established drug pairs as top hits and many more compound pairs with a translational potential. The validity of the tool will hopefully be demonstrated as other scientists start using SynToxProfiler. We hope the screening community will find the tool useful and we are happy to receive feedback on how to improve its options. We foresee SynToxProfiler will allow for more unbiased and systematic means to evaluate the pre-clinical potency of drug combinations toward safe and effective therapeutic applications.

## Methods

### SynToxProfiler workflow

The SynToxProfiler web-application is freely available at https://syntoxprofiler.fimm.fi, together with example drug combination data, video tutorial and step-by-step user instructions. SynToxProfiler enables ranking of drug combinations based on integrated efficacy, synergy and toxicity profiles (Fig 1). Therefore, for each drug combination, SynToxProfiler

first calculates a normalized volume under dose-response surface to quantify combination efficacy based on dose–response measurements on diseased cells, e.g., patient-derived primary cells (see S1 Fig). Then, the combination synergy between each drug pair is estimated using one of the synergy scoring models: Highest Single-Agent [5], Bliss independence [6], Loewe additivity [7], or Zero Interaction Potency (ZIP) [8], as implemented in the SynergyFinder webtool [21]. Normalized volume under the dose-synergy surface is utilized to quantify final combination synergy score (S1 Fig). Next, using the measurements on control cells, if available, the normalized volume under dose–response matrix is calculated to estimate combination toxicity (S1 Fig). Finally, SynToxProfiler ranks the drug combinations based on integrated combination synergy, toxicity and efficacy (STE) score. Alternatively, if measurement on control cells are not available, then the ranking of drug pairs can also be done based merely on combination synergy and efficacy. As a result of the integrative analysis, SynToxProfiler provides a web-based exportable report, which allows users to interactively explore their results (Fig 1 and S2 Fig). An interactive example of web-based report is given at https://syntoxprofiler.fimm.fi/example. A more detailed description of the calculations and workflow with real examples is provided in the technical documentation, https://syntoxprofiler.fimm.fi/howto.

## Calculation of normalized volume

The normalized volume under the dose-response surface is calculated while quantifying combination efficacy and toxicity based on measurements on diseased and control cells, respectively (S1 Fig). Synergy score is calculated based on measurements on diseased cells as normalized volume under synergy matrix (excess matrix of combination responses over expected responses determined by one of the synergy models, such as Bliss). For each combination AB of two drugs, A and B, the normalized volume under the dose-response surface for efficacy ($E_{AB}$), toxicity ($T_{AB}$) and synergy ($S_{AB}$) is calculated as:

$$E_{AB} = \frac{\sum_{x=C_{min}^A}^{C_{max}^A} \sum_{y=C_{min}^B}^{C_{max}^B} E(x,y) \Delta c^A \Delta c^B}{\ln(C_{max}^A / C_{min}^A) \ln(C_{max}^B / C_{min}^B)}. \tag{1}$$

$$T_{AB} = \frac{\sum_{x=C_{min}^A}^{C_{max}^A} \sum_{y=C_{min}^B}^{C_{max}^B} T(x,y) \Delta c^A \Delta c^B}{\ln(C_{max}^A / C_{min}^A) \ln(C_{max}^B / C_{min}^B)}. \tag{2}$$

$$S_{AB} = \frac{\sum_{x=C_{min}^A}^{C_{max}^A} \sum_{y=C_{min}^B}^{C_{max}^B} S(x,y) \Delta c^A \Delta c^B}{\ln(C_{max}^A / C_{min}^A) \ln(C_{max}^B / C_{min}^B)}. \tag{3}$$

Here, $c^A_{min}$ and $c^A_{max}$ are the minimum and maximum tested concentrations of drug A, respectively, and $c^B_{min}$, and $c^B_{max}$ are those of drug B; $\Delta c^A$ and $\Delta c^B$ are the logarithmic increase in concentration of drug A and drug B between two consecutive measurements of dose-response matrix; and $E(x,y)$, $T(x,y)$ and $S(x,y)$ are the efficacy, toxicity and synergy levels respectively at concentration x of drug A and concentration y of drug B. The current approach for volume-based scoring normalizes for the potentially different dose-ranges measured in different drug combinations, as commonly occurring in HTS settings. The extension of formulation for volume -based scoring of synergy, efficacy and toxicity profiles for multi-drug combinations (3 or more drugs) is given in the S1 Text. We have also provided a step-by-step example of the calculation of STE score in a real example case as S4 Fig. The efficacy, toxicity and synergy scores represent the summarized efficacy, toxicity and synergy level for a drug pair summarized over multiple concentration pairs (e.g. 8x8 dose-matrix

design), where each combination is measured in a particular patient sample or cell line. A higher STE score represents higher efficacy, higher synergy and lower toxicity of a drug pair.

## Ranking of drug combinations

SynToxProfiler ranks the drug combinations based on an integrative analysis of synergy, toxicity and efficacy, quantified as STE score (an example is given in S4 Fig). First, the difference in efficacy ($E_{AB}$) and toxicity scores ($T_{AB}$) is calculated for each drug combination to quantify a selective response in diseased cells, relative to that of control cells. We defined this difference as a selective efficacy score ($sE_{AB}$) of a drug combination, calculated by the difference of Eqs (2) and (3). This theoretical concept for selective efficacy has been adopted from the single drug dose-response assays, where the difference in normalized area under the curve (AUC) between diseased and healthy cells is often used to calculate the patient-specific drug efficacies [22, 23]. The final STE score is given by averaging two different ranks of (i) combination synergy score ($S_{AB}$) (the higher is the synergy, the higher is the rank), and (ii) selective combination efficacy ($sE_{AB}$) (the higher is selective efficacy, the higher is the rank):

$$STE_{AB} = \frac{rank(S_{AB}) + rank(sE_{AB})}{2N},\tag{4}$$

where $S_{AB}$ and $sE_{AB}$ are the synergy and selective efficacy scores, respectively, for a combination of drug A and B, calculated using the normalized volume under the dose-response surface; and N is the total number of drug combinations being tested. However, since calculation of STE score using the whole dose-response matrix may miss some of the top hit drug combinations with a narrow synergistic dose window, SynToxProfiler also offers the users a possibility to rank combinations based on the selective efficacy and synergy scores calculated only at the most synergistic area of the drug combination matrix (defined as the 3x3 concentration window with the highest synergy in the dose-response matrix), instead of the default full matrix calculation. The user guide of the tool along with step-by-step example and source code are freely available at https://github.com/IanevskiAleksandr/SynToxProfiler.

## Data submission and reporting

The default input of SynToxProfiler is a text or xlsx file that comprises annotations of each drug combination dose–response matrix, including drug names, concentrations, cell types (e.g. sample or control), and phenotypic responses (e.g. relative inhibition). The number of drug combinations provided in the input file is unrestricted. More information on the input file format is given in the user documentation (https://syntoxprofiler.fimm.fi/howto/). As the result, SynToxProfiler provides an interactive visualization of STE scores using bar charts, as well as 2- and 3-dimensional scatter plots. Publication-quality figures (e.g. heatmap for dose-response and synergy matrix, 2D and 3D scatter plot for different scores) can be exported in PDF files, and all the calculated scores can be downloaded in an xlsx file.

## Ethics statement

The study has been approved by the ethics committee of the University of Helsinki. Peripheral blood mononuclear cells (PBMCs) of a patient with T-cell prolymphocytic leukemia (T-PLL) and a healthy volunteer were used in accordance with the regulations of Finnish Hematological Registry and biobank (FHRB). The written informed consents were obtained from both participants and the study was carried in accordance with the principles of Helsinki declarations.

## Drug combination assay cases

The in-house drug combination testing was carried out at Institute for Molecular Medicine Finland (FIMM). Peripheral blood mononuclear cells (PBMCs) of a patient with T-cell pro-lymphocytic leukemia (T-PLL) and a healthy volunteer were used in accordance with the regulations of Finnish Hematological Registry and biobank (FHRB). The written informed consents were obtained from both participants and the study was carried in accordance with the principles of Helsinki declarations. Twenty combinations of drugs with different mechanisms of actions (see S1 Table) were tested on the PBMCs in 8x8 dose-response matrix assay as described previously [24, 25]. Briefly, drugs were transferred on clear bottom 384-well plates (Corning #3712) using an Echo 550 Liquid Handler (Labcyte) and dissolved in 5μl of Mononuclear Cell Medium (PromoCell, C-28030) dispensed in each well using a MultiFlo FX dispenser (BioTek). After that, 10 000 cells/ 20ul of Mononuclear Cell Medium were dispenced in each well of the plate. The concentration ranges were selected for each compound separately to investigate the full dynamic range of dose-response relationships. After 72 hours incubation at 37˚C and 5% $CO_2$, cell viability of each well was measured using the CellTiter-Glo luminescent assay (Promega) and a Pherastar FS (BMG Labtech) plate reader. As positive (total killing) and negative (non-effective) controls, we used 100 μM benzethonium chloride and 0.1% dimethyl sulfoxide (DMSO), respectively, for calculating the relative efficacy (% inhibition).

The published dataset of 78 antiviral drug combinations was tested at the Integrated Research Facility, National Institutes of Allergy and Infectious Diseases (NIAID), in the Huh7 liver cells infected with Makona isolate, Ebola virus/H.sapiens-tc/GIN/14/WPG-C05, as described in the original study [14] (data available at https://matrix.ncats.nih.gov/matrix-client/rest/matrix/blocks/6323/table and https://matrix.ncats.nih.gov/matrix-client/rest/matrix/blocks/6324/table). Briefly, drugs in 50-μL of Dulbecco's modified Eagle's medium were transferred to the Huh7 cells seeded in black, clear-bottomed, 96-well plates 1 hour prior to inoculation with EBOV/Mak. After 48 hours of viral inoculation, drug combination efficacy was measured in triplicates with a 6 × 6 dose-response matrix design using CellTiter-Glo assays (Promega). The EBOV/Mak virus was detected using mouse antiEBOV VP40 antibody. For the toxicity measurements, the same CellTiter-Glo assay was performed on non–virus-infected Huh7 cells with 3 replicates for each drug concentration, and the assay was repeated at least twice for confirmation. We utilized 77 out of the 78 combinations for the present analysis; colchicine-colchicine pair was removed because the inhibition levels were 100% for all the tested concentrations of this drug combination.

## Supporting information

**S1 Fig. Quantification of efficacy and toxicity in SynToxProfiler.** A schematic representation of calculation of *combination efficacy*, *synergy and toxicity* based on dose–response measurements on diseased cells or control cells. E(x,y) is the response at concentrations x and y of drugs A and B, respectively; $\Delta c^A$ and $\Delta c^B$ are the logarithmic increase in concentration of drug A and drug B between two consecutive measurements of the dose-response matrix. (TIF)

**S2 Fig. Two-dimensional visualization in SynToxProfiler.** Scatter plot showing the distribution of a synergy score (x-axis) and selected efficacy score (y-axis) for 77 combinations tested in the Ebola infected and non–virus-infected Huh7 liver cells. Each drug combination is colored according to its STE score. Users can hover over the combinations to visualize their individual scores (e.g. STE score, or combination synergy, efficacy and toxicity scores), along with

different dose-response matrices (synergy, toxicity, and efficacy), separately for each drug combination as shown here for the apilimod- toremifene citrate combination (right panel). (TIF)

**S3 Fig. The correlation between Bliss synergy scores calculated using SynToxProfiler and Combenefit for full matrix in T-PLL (left panel) and anti-Ebola (right) drug combination screening.** The pearson (R) and Spearman (ρ) correlation coefficients for each data along with respective correlation p-values are shown for both screens. The grey shaded area represents the 95% confidence interval for the fitted regression lines. For calculation of Combenefit synergy score, we have used the SUM_SYN_ANT score. (TIF)

**S4 Fig. A step-by-step example of synergy, efficacy and toxicity (STE) score calculation from dose–response measurements on diseased and control cells.** The user can choose whether the scores are calculated over the full dose-combination matrix, or over the most synergistic 3x3 dose window (the dotted square). (TIF)

**S1 Table. List of drugs used in the assay and their mechanism of action.** (DOCX)

**S2 Table. Comparison of ranks of 20 anti-cancer drug combinations using SynToxProfiler with ZIP, HSA and Bliss synergy models.** The STE score and respective ranks has been calculated for most synergistic area in each combination under ZIP synergy model. (DOCX)

**S3 Table. Comparison of ranks of 77 anti-Ebola drug combinations using SynToxProfiler with ZIP, HSA and Bliss synergy models.** The STE score and respective ranks has been calculated for most synergistic area in each combination under ZIP synergy model. (DOCX)

**S4 Table. Comparison of ranks of T-PLL drug combinations using STE scores from SynToxProfiler and synergy score from Combenefit.** The rank of Bliss synergy and STE scores calculated for full synergy matrix by SynToxProfiler have been compared against SUM_SYN_ANT synergy score from Combenefit. (DOCX)

**S5 Table. Comparison of ranks of 77 anti-Ebola drug combinations using STE scores from SynToxProfiler and synergy score from Combenefit.** The rank of Bliss synergy and STE scores calculated for full synergy matrix by SynToxProfiler have been compared against SUM_SYN_ANT synergy score from Combenefit. (DOCX)

**S1 Data. Summary table for 20 anti-cancer drug combinations measured in 1 T-PLL sample and 1 healthy control analyzed using SynToxProfiler.** (XLSX)

**S2 Data. Summary table for 77 anti-Ebola drug combinations measured in Huh7 cells with and without viral infection and analyzed using SynToxProfiler.** (XLSX)

**S1 Text. Text describing extension of the method for higher-order combinations.** (DOCX)

## Acknowledgments

We thank Prof. Satu Mustjoki for her valuable suggestions about the clinical use of SynTox-Profiler, Prof. Krister Wennerberg for many discussions regarding synergy, toxicity and efficacy scoring approaches for drug combinations, and Andrea Cremaschi for valuable discussions and suggestions on volume-based combination scoring. We thank the T-PLL patients and healthy controls for donating samples for the study.

## Availability and requirement

The SynToxProfiler web-application is publicly available at https://syntoxprofiler.fimm.fi, together with drug combination example data, user instructions, and the source code. The source code is also available at https://github.com/IanevskiAleksandr/SynToxProfiler.

## Author Contributions

**Conceptualization:** Aleksandr Ianevski, Tero Aittokallio, Anil K. Giri.

**Data curation:** Aleksandr Ianevski, Sanna Timonen, Anil K. Giri.

**Formal analysis:** Aleksandr Ianevski, Anil K. Giri.

**Funding acquisition:** Tero Aittokallio.

**Investigation:** Aleksandr Ianevski, Sanna Timonen, Anil K. Giri.

**Methodology:** Aleksandr Ianevski, Sanna Timonen.

**Project administration:** Tero Aittokallio.

**Resources:** Tero Aittokallio.

**Software:** Aleksandr Ianevski.

**Supervision:** Tero Aittokallio, Anil K. Giri.

**Validation:** Sanna Timonen.

**Visualization:** Aleksandr Ianevski, Alexander Kononov.

**Writing – original draft:** Aleksandr Ianevski, Alexander Kononov, Tero Aittokallio, Anil K. Giri.

**Writing – review & editing:** Aleksandr Ianevski, Sanna Timonen, Tero Aittokallio, Anil K. Giri.

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
