## [Decision Letter · Decision Letter 0]

18 Oct 2019

Dear Dr Giri,

Thank you very much for submitting your manuscript, 'SynToxProfiler: interactive analysis of drug combination synergy, toxicity and efficacy', to PLOS Computational Biology. As with all papers submitted to the journal, yours was fully evaluated by the PLOS Computational Biology editorial team, and in this case, by independent peer reviewers. The reviewers appreciated the attention to an important topic but identified some aspects of the manuscript that should be improved.

We would therefore like to ask you to modify the manuscript according to the review recommendations before we can consider your manuscript for acceptance. Your revisions should address the specific points made by each reviewer and we encourage you to respond to particular issues Please note while forming your response, if your article is accepted, you may have the opportunity to make the peer review history publicly available. The record will include editor decision letters (with reviews) and your responses to reviewer comments. If eligible, we will contact you to opt in or out.raised.

- Supporting Information uploaded as separate files, titled 'Dataset', 'Figure', 'Table', 'Text', 'Protocol', 'Audio', or 'Video'.

We hope to receive your revised manuscript within the next 30 days. If you anticipate any delay in its return, we ask that you let us know the expected resubmission date by email at ploscompbiol@plos.org.

Sincerely,

Manja Marz

Software Editor

PLOS Computational Biology

[LINK]

Reviewer's Responses to Questions

**Comments to the Authors:**

Reviewer #1: Comments: A web-based tool is presented that allows the calculation of drug combination synergy, toxicity and efficacy. The justification for the investigation is that most methods that evaluate drug combinations do so based on a synergy parameter, such as the Bliss independence value, rather than an integrated metric that combines drug toxicity (need control cell data) and efficacy and synergy. Overall, this is a nice addition to the drug combination screening literature; however, there are a number of points that need to be addressed.

- It is hard to follow all the mathematical steps to attain an STE. They need to show all the equations for EAB, TAB and STEAB and use real numbers. Equation 1 appears to be the basis of the calculations, but a step-by-step outline is required. Further, how does the choic of synergistic model effect the STE calculation?

- The website is quite reasonable, but it could be improved with more detailed directions. For example, step 1 might be how to create the excel file and comment on the need to specify units and format. One can find this information on the website but I feel it could be more clearly presented.

- It would also be interesting to compare data analyses pf SynTox Profiler with programs that only do synergy calculations. The benefit would be 2-fold. One, one could compare the synergy values using the same model (Bliss, etc) with 2 software programs (do they agree or not and why not if that’s the case?). Two, the complete STE parameters could then be compared and rated as to the added value of their approach. It looks like they did this within the SynTox Profiler software. One program that might consider to compare to is Combenefit (Bioinformatics, 32(18), 2016, 2866–2868).

- Can they provide insights – after using the program on multiple datasets – on drug combinations that show positive synergy, but fail the STE?

Reviewer #2: The article by Giri and coworkers introduces SynToxProfiler as a tool to help researches investigate drug combinations by taking into account not only synergy/efficacy, but also toxicity. Overall I like the idea although I have a some minor reservations which I outline below.

1. An issue with this approach is that the toxicity predictions are only as good as the experimental method used. That is to say that in vitro methods to measure toxicity may not capture clinical toxicity for numerous reasons (eg limited tissue types tested or ADME effects not taken into account). The authors should briefly mention this limitation in the discussion.

2. Related to the comment above, the introduction would also benefit with a short explanation saying that toxicity is measured on non-diseased cells.

3. It would be helpful to more clearly define (methods section) what the efficacy, toxicity, synergy scores are. Are the efficacy and toxicity scores just the volumes under the dose-response surfaces? What is the synergy score. I have no feeling for what the numbers represent. I recommend including in the introduction a few extra sentences describing how synergy, toxicity and efficacy were calculated and going into significantly more detail in the methods section.

4. Is volume under the dose response curve an appropriate metric for measuring efficacy of a combination - does this ignore the shape of the dose response surface? For example I can imagine a situation in which when drugs are combined at a certain ratio they are particularly toxic, but at a different ratio they are not toxic. The latter ratio might be therapeutically useful, but this could be missed if only volume is considered because this would be an “average” of the toxic and non-toxic ratios. Please consider whether it is appropriate to include this in the discussion – possibly with a recommendation to users to look at individual response surfaces and not just the overall score when selecting drug combinations to investigate?

5. I think that the method of validating the SynToxProfiler is a little weak – in essence by showing that it predicts a few successful clinical combinations. However, I can think of no better way to validate it. Perhaps the authors could add a sentence in the discussion saying that the validity of the tool will hopefully be demonstrated as other scientists start to use it?

6. The figure numbering and order is mixed up.

7. Define ZIP abbreviation

8. Define colour coding fig 1A

9. Fig 1 seems to suggest a good toxicity score is good – I am struggling with this if it’s the volume under the toxicity curve.

**Have all data underlying the figures and results presented in the manuscript been provided?**

Reviewer #1: Yes

Reviewer #2: Yes

PLOS authors have the option to publish the peer review history of their article (what does this mean?). If published, this will include your full peer review and any attached files.

Reviewer #1: No

Reviewer #2: Yes: Alan Richardson

---

## [Decision Letter · Decision Letter 1]

11 Dec 2019

Dear Dr Giri,

We are pleased to inform you that your manuscript 'SynToxProfiler: interactive analysis of drug combination synergy, toxicity and efficacy' has been provisionally accepted for publication in PLOS Computational Biology.

In the meantime, please log into Editorial Manager at https://www.editorialmanager.com/pcompbiol/, click the "Update My Information" link at the top of the page, and update your user information to ensure an efficient production and billing process.

One of the goals of PLOS is to make science accessible to educators and the public. PLOS staff issue occasional press releases and make early versions of PLOS Computational Biology articles available to science writers and journalists. PLOS staff also collaborate with Communication and Public Information Offices and would be happy to work with the relevant people at your institution or funding agency. If your institution or funding agency is interested in promoting your findings, please ask them to coordinate their releases with PLOS (contact ploscompbiol@plos.org).

Thank you again for supporting Open Access publishing. We look forward to publishing your paper in PLOS Computational Biology.

Sincerely,

Manja Marz

Software Editor

PLOS Computational Biology

Reviewer's Responses to Questions

**Comments to the Authors:**

Reviewer #1: All comments appropriately addressed to improve the paper.

Reviewer #2: Thank you for considering the recommendations, I think the manuscript is now very significantly improved.

**Have all data underlying the figures and results presented in the manuscript been provided?**

Reviewer #1: None

Reviewer #2: Yes

PLOS authors have the option to publish the peer review history of their article (what does this mean?). If published, this will include your full peer review and any attached files.

Reviewer #1: No

Reviewer #2: Yes: Alan Richardson

---

## [Editor Report · Acceptance letter]

21 Jan 2020

PCOMPBIOL-D-19-01247R1 

SynToxProfiler: An interactive analysis of drug combination synergy, toxicity and efficacy

Dear Dr Giri,

I am pleased to inform you that your manuscript has been formally accepted for publication in PLOS Computational Biology. Your manuscript is now with our production department and you will be notified of the publication date in due course.

With kind regards,

Laura Mallard
